# Hyperosmotic stress induces epithelial-mesenchymal transition through rearrangements of focal adhesions in tubular epithelial cells

**Takashi Miyano**⬤*, **Atsushi Suzuki, Naoya Sakamoto**⬤*

Department of Mechanical Systems Engineering, Graduate School of Systems Design, Tokyo Metropolitan University, Tokyo, Japan

* miyano-takashi@ed.tmu.ac.jp (TM); sakan@tmu.ac.jp (NS)

**Data Availability Statement:** All relevant data are within the manuscript and its S1–S3 Figs and S1 File and S1 Raw images.

## Abstract

Epithelial-mesenchymal transition (EMT) of tubular epithelial cells is a hallmark of renal tubulointerstitial fibrosis and is associated with chronic renal injury as well as acute renal injury. As one of the incidences and risk factors for acute renal injury, increasing the osmolality in the proximal tubular fluid by administration of intravenous mannitol has been reported, but the detailed mechanisms remain unclear. Hyperosmotic conditions caused by mannitol in the tubular tissue may generate not only osmotic but also mechanical stresses, which are known to be able to induce EMT in epithelial cells, thereby contributing to renal injury. Herein, we investigate the effect of hyperosmolarity on EMT in tubular epithelial cells. Normal rat kidney (NRK)-52E cells were exposed to mannitol-induced hyperosmotic stress. Consequently, the hyperosmotic stress led to a reduced expression of the epithelial marker E-cadherin and an enhanced expression of the mesenchymal marker, α-smooth muscle actin (α-SMA), which indicates an initiation of EMT in NKR-52E cells. The hyperosmotic condition also induced time-dependent disassembly and rearrangements of focal adhesions (FAs) concomitant with changes in actin cytoskeleton. Moreover, prevention of FAs rearrangements by cotreatment with Y-27632, a Rho-associated protein kinase inhibitor, could abolish the effects of hyperosmotic mannitol treatment, thus attenuating the expression of α-SMA to the level in nontreated cells. These results suggest that hyperosmotic stress may induce EMT through FAs rearrangement in proximal tubular epithelial cells.

## Introduction

Epithelial-mesenchymal transition (EMT) is a widely accepted mechanism by which injured tubular epithelial cells transform into myofibroblasts, and is involved in the pathogenesis of not only chronic kidney diseases but also in acute kidney injury (AKI) [1, 2]. During EMT, the tubular epithelial cells lose their epithelial characteristics and acquire mesenchymal features, concomitant with the downregulation of epithelial markers, including E-cadherin, and the

**Funding:** This study was supported in part by Grants-in-Aid for Scientific Research by the MEXT of Japan (No. 17H0277, 18H03521, and 18K19934). The funders had no role in study design, data collection and analysis, decision to publish, or preparation of the manuscript.

**Competing interests:** The authors have declared that no competing interests exist.

upregulation of mesenchymal markers, including α-smooth muscle actin (α-SMA) and vimentin [3]. α-SMA-positive myofibroblasts are known to induce the expression of profibrotic factors such as collagen, fibronectin, and plasminogen activator inhibitor type 1 (PAI-1) [4–6]. In fact, a very small number of interstitial α-SMA-positive myofibroblasts has been shown to arise from EMT *in vivo* model [7]. However, many inhibitors and small molecules against EMT exerted profound therapeutic effects on renal diseases by suppressing differentiation into myofibroblasts and production of extracellular matrix [8, 9]. Therefore, although the proportion of EMT-derived myofibroblasts is small, the EMT program is believed to play an important role in the progression of renal disease.

Elevation of blood plasma osmolality by administration of osmolytes, such as mannitol, is clinically performed to reduce intracranial and intraocular pressures, but excessive administration is known to be an incidence and risk factor for AKI [10–12]. Mannitol-induced AKI is characterized by the structural changes that occur at the cellular level in the proximal tubule, including intracytoplasmic vacuolization and swelling of cells [13–15]. The proximal tubule is the first segment of the kidney tubule, and excessive mannitol is not reabsorbed and leads to increase the osmolality in the proximal tubular fluid. The exact mechanism of mannitol-induced AKI has not been clarified yet but changes in tubular osmolarity and the osmolar gap may be the contributing factors [15]. Besides, previous studies have reported that interstitial α-SMA-positive myofibroblasts appear around the proximal tubules in the AKI model [16, 17]. Therefore, hyperosmotic stress could be involved in the induction of EMT.

EMT occurs in response to numerous factors such as cytokines, hormones, and autacoids [3, 18]. Interestingly, recent studies have demonstrated that mechanical stresses such as fluid shear stress and cyclic stretch could also induce EMT [19, 20]. Changes in osmotic conditions of proximal tubular epithelial cells can generate not only osmotic but also mechanical stresses in cells by altering cell volume and cytoplasm membrane tension [21]. Cells sense the mechanical stresses through cell–extracellular matrix (ECM) adhesion structures known as focal adhesions (FAs), which are connected to the actin structure through FA-associated binding proteins [22] and transmit force-induced signals between the ECM and cytoskeleton [23–25]. The cytoskeleton and FAs are not static structures that simply transmit force, but they are always dynamically reorganized [26]. It is also known that the induction of EMT causes a dynamic remodeling of actin cytoskeleton from the cortical organization of actin filaments to thick actin stress fibers, which is a hallmark of mesenchymal cells [27, 28]. Although it has been reported that hyperosmotic stress induces depolymerization and rearrangement of actin filaments [29], to the best of our knowledge, there are no studies regarding the effects of hyperosmolarity on FAs. In fact, it has been proposed that the size of FAs controls the recruitment of α-SMA to actin stress fibers [30–32]. Therefore, FAs could be involved in the induction of EMT by controlling the expression of α-SMA.

This study investigated the mechanisms involved in the hyperosmolarity-induced EMT. For this purpose, we examined the effects of hyperosmotic stress on α-SMA expression and FA dynamics in rat tubular epithelial cells. Our results demonstrated that hyperosmotic stress caused dynamic changes in the cytoskeleton characterized by disassembly and rearrangements of FAs and that preventing FA rearrangements could suppress the hyperosmotic stress-induced differentiation into α-SMA-positive myofibroblasts.

## Materials and methods

### Cell lines and reagents

Normal rat kidney cells (NRK-52E), a tubular epithelial cell line derived from rat kidney, were directly purchased from the Japanese Collection of Research Bioresources (JCRB No. IFO50480, Osaka, Japan). Both mannitol and urea for adjusting the osmolarity in the medium

were purchased from FUJIFILM Wako Pure Chemical (Osaka, Japan). Y-27632, ROCK (Rho-associated protein kinase) inhibitor, was also obtained from FUJIFILM Wako Pure Chemical. Antibodies for α-SMA, vinculin, E-cadherin, and GAPDH were purchased from DAKO (Glostrup, Denmark), Invitrogen (Carlsbad, CA), abcam (Cambridge, MA), and Cell Signaling Technology (Danvers, MA), respectively.

## Cell culture and hyperosmotic stimulation

NRK-52E cells were maintained in Dulbecco's modified Eagle's medium (DMEM; FUJIFILM Wako Pure Chemical) supplemented with 10% fetal bovine serum (FBS; Cosmo Bio, Tokyo, Japan), 1 U/mL penicillin–streptomycin (FUJIFILM Wako Pure Chemical) and cultured at 37˚C in 5% $CO_2$ atmosphere. For providing hyperosmotic stress to the cells, the medium was switched to DMEM supplemented with 0.5% FBS and cultured for 24 h and then to a hyperosmotic medium containing mannitol or urea. Cells reaching about 80% confluence were used for the experiments. The hyperosmotic medium was prepared by adding mannitol or urea to DMEM (0 mM: 313 ± 1.5 mOsmol/L, 100 mM: 428 ± 2.4 mOsmol/L, 200 mM: 531 ± 4.8 mOsmol/L (mean ± S.E.)). The dose of 200 mM mannitol (amount to 36 g/L) used in the present study exceeded the plasma mannitol concentration (29 g/L) after administration high dose of mannitol (4 g/kg/h) [10].

## RNA extraction and quantitative real-time PCR

NRK-52E cells were lysed in ISOGEN (NIPPON GENE, Toyama, Japan), and to obtain cDNA, the reverse transcription reaction was performed using the ReverTra Ace qPCR RT Master Mix (TOYOBO, Osaka, Japan). Quantification of cDNA was performed using THUNDERBIRD SYBR qPCR Mix (TOYOBO) and the Thermal Cycler Dice Real-Time System Tp800 (TaKaRa Biomedicals, Shiga, Japan). PCR was conducted in 5 μM of cDNA, 10 μM of master mix, and 5 pM of sense, and antisense primers. Table 1 shows the primer sets used for PCR. The relative mRNA expression levels of the target genes in each sample were calculated as the CT value, which is the cycle number at which the fluorescence signal is greater than a defined threshold. The expression of each gene was normalized with the housekeeping gene GAPDH, and the relative mRNA levels were analyzed by the ΔΔCT method and compared to those of untreated, time-matched control samples.

## Immunofluorescence staining

After the stimulation of hyperosmotic stress with mannitol or urea, the morphology of NRK-52E cells was observed under the phase-contrast microscope (FSX-100, Olympus, Tokyo,

**Table 1. Primers used in this study.**

| Gene Name | Sense | Antisense |
|---|---|---|
| Rat GAPDH | TGACAACTTTGGCATCGTGG | GGGCCATCCACAGTCTTCTG |
| Rat SNAIL | CAGATGGCTGATGGAAGGCA | CAGCTGTGTCCAGAGGCTAC |
| Rat TWIST | AGAGATTCCCAGAGGCAACG | TGACTGATTGGCAAGACCTC |
| Rat Collagen-I | ACTGGTACATCAGCCCAAAC | GGAACCTTCGCTTCCATACTC |
| Rat PAI-1 | GACAATGGAAGAGCAACATG | ACCTCGATCTTGACCTTTTG |
| Rat Fibronectin | GTGATCTACGAGGGACAGC | GCTGGTGGTGAAGTCAAAG |
| Rat E-cadherin | GAGGTCTTTGAGGGATCTGTTG | GGCAGCATTGTAGGTGTTTATG |
| Rat vimentin | CTTCCCTGAACCTGAGAGAAAC | GTCTCTGGTTTCAACCGTCTTA |
| Rat α-SMA | AGGGAGTGATGGTTGGAATG | GGTGATGATGCCGTGTTCTA |

Japan). For immunofluorescence staining, cells were fixed with 4% paraformaldehyde for 15 min and then permeabilized with 0.5% Triton X-100 for 15 min at room temperature. After blocking with 3% BSA (Sigma-Aldrich) in PBS for 1 h, cells were incubated with E-cadherin, α-SMA, or vinculin antibody (diluted to 1:200 in the blocking solution) for 1 h at room temperature, followed by staining with Alexa Fluor 488- or 546-conjugated secondary antibody (1:500, Invitrogen) for 1 h. For staining the nuclei and F-actin, cells were also incubated with Hoechst 33342 (Invitrogen) and Alexa Fluor 546-conjugated phalloidin (Invitrogen), respectively, for 15 min. All samples of immunofluorescence staining were imaged by confocal microscopy (FV3000, Olympus).

## Image processing

The ImageJ Fiji software (version 1.52b, NIH) was used for the processing of images. After removing the background, the cell outline was detected and binarized. In the region excluding the cell edge, a vinculin dot of $\geq 0.8$ $\mu m^2$ was defined as FA [33], and "the number of FAs per cell" and "the area of each FA" were obtained for each cell. Projected cell areas were measured by manually outlining the cells on fluorescent images of F-actin (S1 Fig). Relative immunofluorescence intensities of E-cadherin and α-SMA were quantified by using the ImageJ Fiji [34].

## Western blotting

NRK-52E cells were lysed in a lysis buffer (150 mM NaCl, 1 mM EDTA, 50 mM Tris, 1% NP-40, 1% Triton X-100, and 1% protease inhibitor cocktail (Sigma-Aldrich, Kyoto, Japan)) and centrifuged to collect the supernatant. The loading samples containing "1" to "5" μg of protein were electrophoresed on SDS-PAGE gels and transferred to polyvinylidene difluoride (PVDF) membranes. After blocking the PVDF membranes with 5% nonfat milk in TBST (Tris-buffered saline (TBS) containing 0.1% Tween 20) for 1 h, the membranes were incubated with E-cadherin, α-SMA or GAPDH, diluted up to 1:2000, at 4˚C overnight. Goat anti-rabbit IgG, horseradish peroxidase-linked secondary antibody (Cell Signaling Technology), diluted up to 1:2000 was detected by adding enhanced chemiluminescent reagent. Blots were stripped with Restore™ Western Stripping Buffer (Thermo Fisher Scientific) and re-probed with different antibodies. The band intensity was quantified from scanned membrane images using the ImageJ Fiji software.

## Statistical analysis

All experiments were repeated at least three times. All results were analyzed using the GraphPad Prism 5 ver. 5.0 software (GraphPad Software) and expressed as mean ± standard error (S. E.) of at least three separate experiments. Comparison of two groups was performed using Student's $t$-test. For multiple group comparisons, one-way analysis of variance (ANOVA), followed by Steel, Steel–Dwass, Dunnett's and Tukey's post hoc test was used to compare groups with nonparametric and parametric data, respectively. $P < 0.05$ was considered to represent a statistically significant difference.

# Results

## Hyperosmolarity induces the EMT of NRK-52E cells

The effect of hyperosmotic stress induced by mannitol on EMT in NRK-52E cells was examined by immunofluorescence staining (Fig 1A). In the absence of extracellular osmolytes, E-cadherin, epithelial cell marker, was abundantly localized to the plasma membrane. The expression of α-SMA, a mesenchymal cell marker was rarely detected, which is consistent with

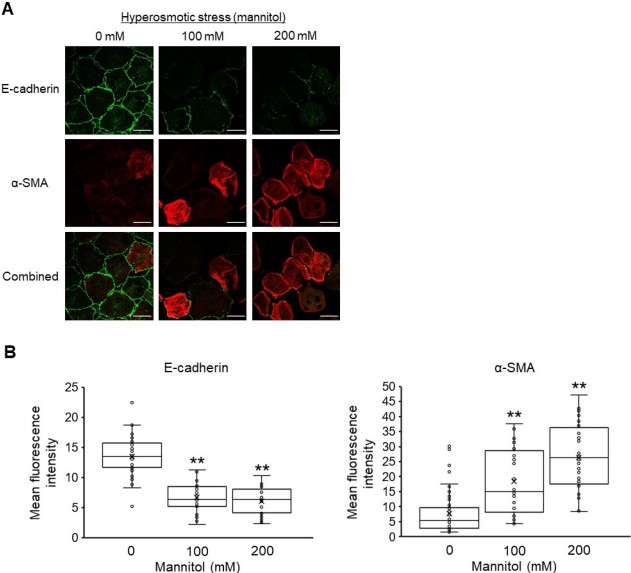

**Fig 1. Effects of hyperosmotic mannitol stress on the epithelial-mesenchymal transition (EMT) of NRK-52E cells.**
(A) Typical fluorescence images of E-cadherin (green), α-SMA (red), and combined (green and red) cultured with 0, 100, and 200 mM mannitol for 12 h. Bar, 25 μm. (B) Quantitation of the changes in the mean fluorescence intensity of E-cadherin and α-SMA (n = 46 from 0 mM, n = 30 from 100 mM, n = 26 from 200 mM) by immunofluorescence staining. Data are presented as box and whisker plots with average (×), median, IQR, and minimum and maximum values. The n indicates the number of independent experiments. **P < 0.01 from the data of 0 mM (Dunnett's test).

previous studies investigating the induction of EMT using NRK-52E cells [35–37]. When the cells were stimulated by hyperosmotic stress for 12 h, the expression of E-cadherin was markedly decreased, and the expression of α-SMA was dramatically increased in the cytoplasm in a mannitol-dose-dependent manner (Fig 1A). Quantitatively, the mean fluorescence intensity of E-cadherin and α-SMA was significantly decreased and increased, respectively, by hyperosmotic mannitol stress (Fig 1B). These changes in expression are consistent with the features of EMT.

Since E-cadherin downregulation is an initial hallmark of the differentiation of epithelial cells to the mesenchymal phenotype [38, 39], we measured the expression levels of Snail and Twist, which are the major transcription factors that regulate E-cadherin expression [3, 18, 40], to verify the mechanism underlying the hyperosmotic stress-induced E-cadherin downregulation. When cells were stimulated by hyperosmotic stress for 12 h, the mRNA expression levels of Snail and Twist were increased in a mannitol-dose-dependent manner (Fig 2). These data suggest that hyperosmotic mannitol stress induces EMT in NRK-52E cells with a reduction of E-cadherin expression by the upregulation of both Snail and Twist.

We also explored the E-cadherin and α-SMA expression of NRK-52E cells under urea-mediated hyperosmolarity to further understand the effect of hyperosmotic stress-induced cell volumetric changes on EMT. Previous studies have reported that when proximal tubules were exposed to hyperosmotic urea, the reduction in cell volume was markedly smaller than that under hyperosmotic mannitol treatment [41, 42]. Unlike the result obtained with mannitol, treatment with hyperosmotic urea resulted in no changes in the expression of E-cadherin and α-SMA compared with the control (0 mM) (Fig 3A and 3B). We also found by quantifying projected cell areas that the treatment with 200 mM mannitol resulted in a temporary decrease in cell areas at 0.25 and 0.5 h, which recovered at 2 h to the initial level (Fig 4A), but the urea treatment did not cause significant changes (Fig 4B). These results suggest that the

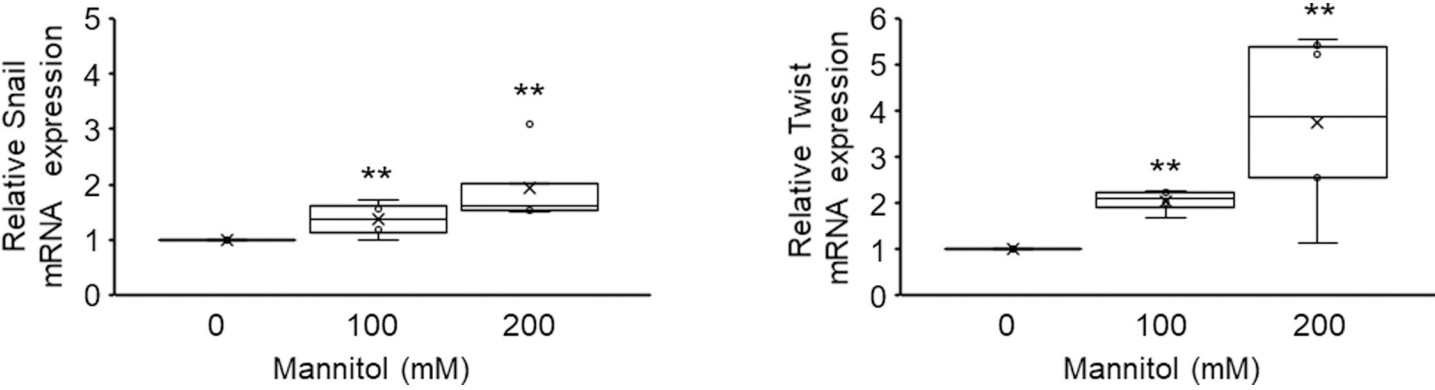

**Fig 2. Effects of hyperosmotic mannitol stress on the expression of Snail and Twist.** NRK-52E cells were treated with mannitol (0, 100, 200 mM) for 12 h, and mRNA expression was analyzed by real-time PCR. Quantitation of the changes in the mRNA expression of Snail (n = 6 from 0 mM, n = 4 from 100 mM, n = 5 from 200 mM) and Twist (n = 7 from 0 mM, n = 4 from 100 mM, n = 6 from 200 mM) by real-time PCR. Relative gene expression levels were calculated considering mannitol (0 mM) as 1 and plotted. Data are presented as box and whisker plots with average (×), median, IQR, and minimum and maximum values. The n indicates the number of independent experiments. **P < 0.01 from the data of 0 mM (Steel test).

hyperosmotic stress-induced EMT in NRK-52E cells requires cell shrinkage concomitant with the osmotic gradient between the intracellular and extracellular compartments as a result of water efflux from the cells.

## Hyperosmolarity induces the dynamic changes in FAs

Based on the above-described results, we hypothesized that the cell shrinkage caused by hyper-osmotic stress interferes with the structure and organization of the cytoskeletal network by

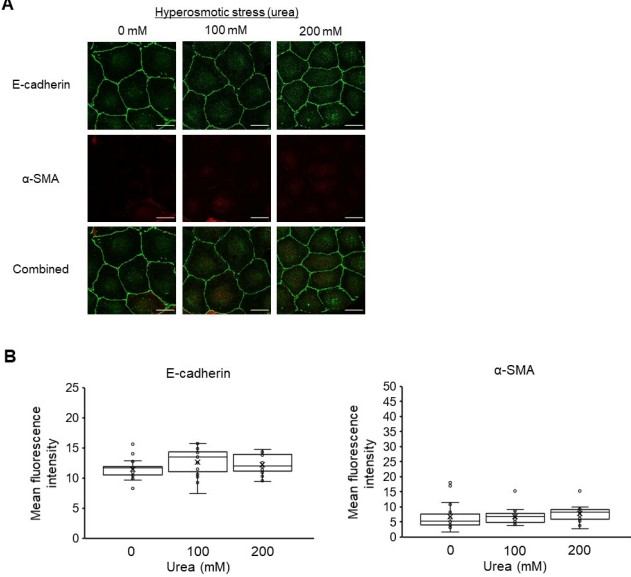

**Fig 3. Effects of hyperosmotic urea stress on the epithelial-mesenchymal transition (EMT) of NRK-52E cells.** (A) Typical fluorescence images of E-cadherin (green), α-SMA (red), and combined (green and red) cultured with 0, 100, and 200 mM urea for 12 h. Bar, 25 μm. (B) Quantitation of the changes in the mean fluorescence intensities of E-cadherin and α-SMA (n = 19 from 0 mM, n = 19 from 100 mM, n = 19 from 200 mM) by immunofluorescence staining. Data are presented as box and whisker plots with average (×), median, IQR, and minimum and maximum values. The n indicates the number of independent experiments.

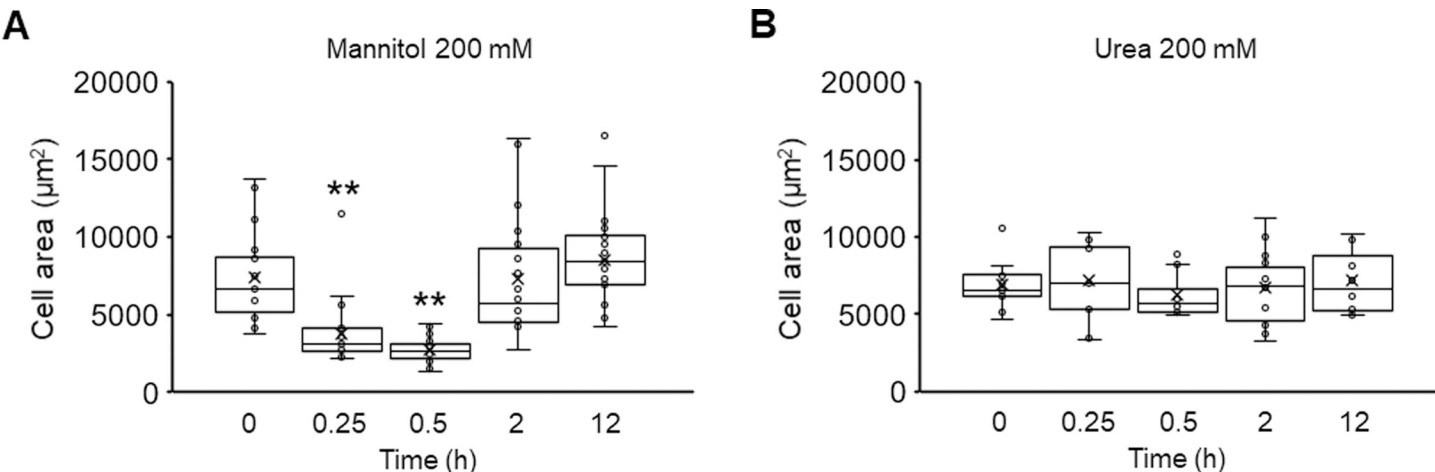

**Fig 4. Effects of hyperosmolarity on cell area of NRK-52E cells.** (A, B) NRK-52E cells were treated with mannitol (200 mM) (A) or urea (200 mM) (B) for 0, 0.25, 0.5, 2, and 12 h (mannitol; 0 h (n = 25), 0.25 h (n = 26), 0.5 h (n = 28), 2 h (n = 22), 12 h (n = 29): urea; 0 h (n = 12), 0.25 h (n = 13), 0.5 h (n = 12), 2 h (n = 14), 12 h (n = 12)). Data are presented as box and whisker plots with average (×), median, IQR and minimum and maximum values. The n indicates the number of cells analyzed. ** P < 0.01 from the data of 0 h (Dunnett's test).

limiting the intracellular space. To test this hypothesis, we focused on the effect of hyperosmotic stress on actin cytoskeletal structures, which is known to be responsible for mediating various important cellular processes such as cell structural support and functional regulation [29]. Although we observed no change in the actin cytoskeleton in NRK-52E cells cultured with 100 mM mannitol (Fig 5A), the actin cytoskeletal structure exhibited drastic time-dependent changes after being treated with 200 mM mannitol (Fig 5B); the actin filaments were disassembled by treatment with 200 mM mannitol for 0.5 h and then reorganized into thick stress fibers at 12 h. These results are consistent with those of previous studies that demonstrated hyperosmolarity-induced depolymerization and reorganization of the actin cytoskeleton for adaptive responses [29].

We further evaluated the effect of hyperosmotic treatment on actin cytoskeletal structures by evaluating the rearrangement of FAs, which play important roles in the organization and structure of actin filaments. Mannitol-treated cells were examined for their contents of vinculin, FAs marker, by fluorescence immunostaining. Like the actin cytoskeletal structures, treatment with 100 mM mannitol did not cause any detectable changes in the vinculin fluorescence distribution pattern (Fig 5A), which was similar to control cells demonstrating widespread labeling of FAs (0 h in Fig 5A). In contrast, in the cells treated with 200 mM mannitol, vinculins were disassembled in the central region and almost localized along the edges of cells after 30 min and then recovered to a similar distribution pattern as that of control cells at 12 h (Fig 5B). To quantify the effect of hyperosmotic stress treatment on FA changes, we calculated "the number of FAs per cell" and "the area of each FA." Upon treatment with 200 mM mannitol, there was a significant decrease in the number of FAs after 0.25 and 0.5 h (0 h, 23.7 ± 2.0; 0.25 h, 4.7 ± 0.6 ($P < 0.01$); 0.5 h, 3.6 ± 0.6 ($P < 0.01$)) (Fig 5C), which recovered to the control level at 2 h (21.1 ± 3.8) and then significantly increased at 12 h compared with the initial value (46.9 ± 9.0 ($P < 0.01$)) (Fig 5C). The area of each FA was similar to the number of FAs. Within the first 0.5 h of 200 mM mannitol treatment, each FA area gradually decreased (0 h, 2.3 ± 0.1 $\mu m^2$; 0.25 h, 1.8 ± 0.2 $\mu m^2$; 0.5 h, 1.2 ± 0.1 $\mu m^2$ ($P < 0.01$)) (Fig 5D). The value of FA area returned to the level of the untreated cells at 2 h (2.2 ± 0.1 $\mu m^2$) and then significantly increased at 12 h (3.0 ± 0.2 $\mu m^2$ ($P < 0.01$)) (Fig 5D). Treatment with 100 mM mannitol resulted in no significant changes in both FA number and FA area (Fig 5A, 5C and 5D). These

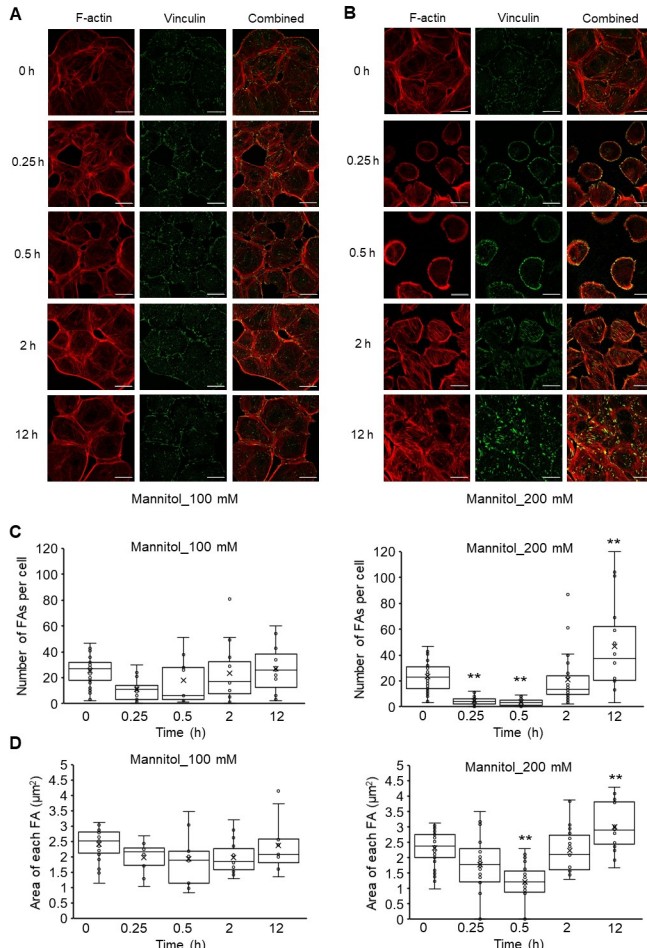

**Fig 5. Effects of hyperosmotic mannitol stress on actin and vinculin distribution in NRK-52E cells.** (A, B) Cells were cultured with 100 (A) or 200 mM (B) mannitol for 0, 0.25, 0.5, 2, and 12 h. Typical fluorescence images of F-actin (red), vinculin (green), and combined (red and green). Bar, 25 μm. (C, D) A summary of vinculin staining data from (A) and (B), showing the number of FAs per cell (C) and area of each FA (D) (100 mM; 0 h (n = 25), 0.25 h (n = 13), 0.5 h (n = 9), 2 h (n = 15), 12 h (n = 11): 200 mM; 0 h (n = 33), 0.25 h (n = 28), 0.5 h (n = 27), 2 h (n = 28), 12 h (n = 16)). Data are presented as box and whisker plots with average (×), median, IQR and minimum and maximum values. The n indicates the number of cells analyzed. $^{**}P < 0.01$ from the data of 0 h (Dunnett's test).

findings clearly demonstrate that hyperosmotic condition induces the disassembly and subsequent rearrangements of FAs concomitant with actin filament dynamics change in NRK-52E cells.

## Hyperosmolarity promotes the incorporation of α-SMA into actin stress fibers

We observed that treatment with 200 mM mannitol for 12 h significantly increased the area of each FA in NRK-52E cells. Based on the findings of previous studies, FA size controls the recruitment of α-SMA to actin stress fibers [30], we further hypothesized that hyperosmotic stress promotes the incorporation of α-SMA into actin stress fibers. Strikingly, after treatment with 200 mM mannitol for 12 h, the expression level of α-SMA protein in the cells was significantly increased (Fig 6A), and the association of α-SMA with actin filaments increased cellular anisotropy compared with the control (0 mM) (Fig 6B). We also found the decreased

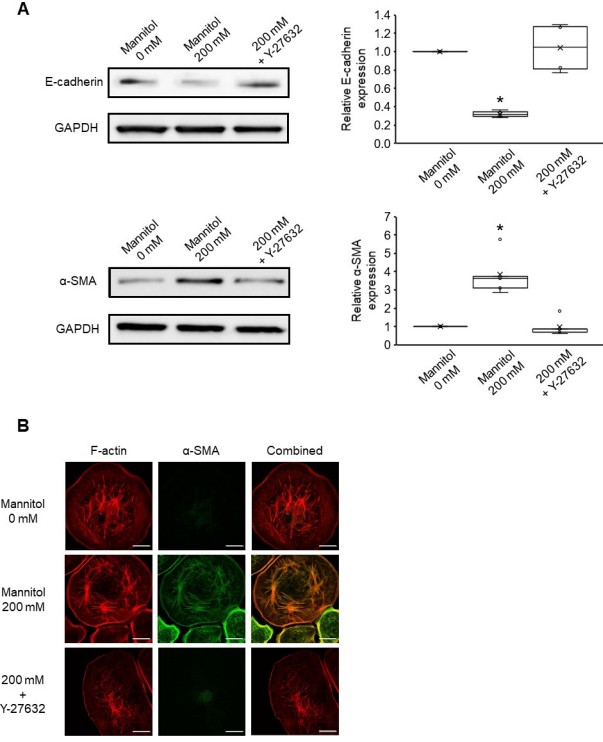

**Fig 6. Effects of FAs on the recruitment of α-SMA to actin stress fibers.** NRK-52E cells were treated with mannitol (200 mM) or cotreated with Y-27632 (1 μM) for 12 h. (A) Western blot analysis for E-cadherin (n = 4 from mannitol (0 mM), n = 4 from mannitol (200 mM), n = 4 from mannitol (200 mM) + Y-27632) and α-SMA (n = 5 from mannitol (0 mM), n = 5 from mannitol (200 mM), n = 5 from mannitol (200 mM) + Y-27632). Left: The representative bands obtained Western blotting. Right: Quantification analysis of E-cadherin and α-SMA. Relative expression was calculated as normalized to untreated cells (mannitol 0 mM). Data are presented as box and whisker plots with average (×), median, IQR, and minimum and maximum values. The n indicates the number of independent experiments. *P < 0.05 from the data of 0 mM (Steel–Dwass test). (B) Typical fluorescence images of F-actin (red), α-SMA (green), and combined (red and green). Bar, 25 μm.

expression of E-cadherin at the same time, which is consistent with the features of EMT (Fig 6A). These observations indicate that hyperosmotic condition could promote the expression and recruitment of α-SMA to the actin stress fibers in NRK-52E cells.

## Cotreatment with Y-27632 suppresses hyperosmolarity-induced FA rearrangements and colocalization of F-actin and α-SMA

To investigate whether FA rearrangements directly affect the hyperosmolarity-induced EMT in NRK-52E cells, we explored the experimental conditions that prevent the rearrangement of FAs. We observed that treatment with 1 μM Y-27632, a ROCK inhibitor, could suppress FA rearrangements induced by 200 mM mannitol treatment, but it had no appreciable effect on actin filament reorganization (Fig 7A). Quantitatively, the average of FA area of cells cotreated with Y-27632 and mannitol exhibited a significant decrease until 0.5 h (0 h, 2.6 ± 0.1 μm$^2$; 0.25 h, 1.3 ± 0.3 μm$^2$ ($P < 0.01$); 0.5 h, 1.4 ± 0.1 μm$^2$ ($P < 0.01$)) (Fig 7B). Even after 2 h since the beginning of the cotreatment, the significant decrease in the value of FA area continued at 2 and 12 h (1.6 ± 0.2 μm$^2$ ($P < 0.01$) and 1.9 ± 0.1 μm$^2$ ($P < 0.05$), respectively) (Fig 7B). The values of FA area at 2 and 12 h after cotreatment with Y-27632 were smaller than those obtained under treatment with 200 mM mannitol alone (Fig 5D) at the same time points; this result was statistically significant (Fig 7C). These results demonstrated that cotreatment with 200 mM

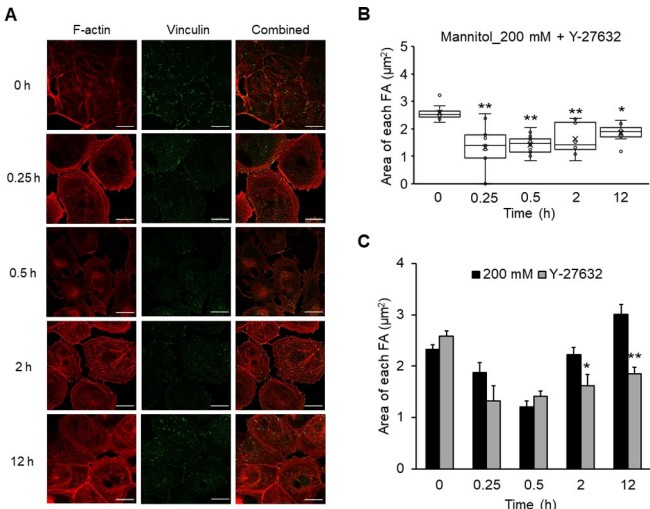

**Fig 7. Experimental conditions that prevent the rearrangement of FAs without affecting actin cytoskeleton reorganization.** NRK-52E cells were cotreated with mannitol (200 mM) and Y-27632 (1 μM) for 0, 0.25, 0.5, 2, and 12 h. (A) Typical fluorescence images of F-actin (red), vinculin (green), and combined (red and green). Bar, 25 μm. (B) A summary of vinculin staining data from (A), showing the area of each FA (mannitol (200 mM) + Y-27632; 0 h (n = 9), 0.25 h (n = 9), 0.5 h (n = 14), 2 h (n = 8), 12 h (n = 8)). Data are presented as box and whisker plots with average (×), median, IQR, and minimum and maximum values. The n indicates the number of cells analyzed. (C) A comparison between mannitol (200 mM) alone and cotreated with Y-27632 (1 μM) from (B). The data of 200 mM mannitol alone (black bars) were identical to those in Fig 5D, which were shown for comparisons. Data are mean ± SEM. $^*P < 0.05$, $^{**}P < 0.01$ from the data of mannitol (200 mM) at the same time point (Student's $t$-test).

mannitol and 1 μM Y-27632 was an experimental condition that could prevent hyperosmotic stress-induced FA rearrangements without affecting actin cytoskeletal dynamics in NRK-52E cells.

We then investigated the effects of treatment with 1 μM Y-27632, which prevented the hyperosmolarity-induced FA rearrangements, on the recruitment of α-SMA to actin stress fibers. Preventing the FA rearrangements attenuated the effects of hyperosmolarity. Even under the hyperosmotic conditions, treatment with Y-27632 significantly reduced the colocalization of F-actin and α-SMA fluorescence signals similar to the control levels (Fig 6B). These results suggest that the FA rearrangements induced by hyperosmotic stress play an important role in the expression of α-SMA and its incorporation into actin stress fibers in NRK-52E cells. Furthermore, using Western blotting and immunofluorescence staining, we found that the Y-27632 treatment abolished both the increase in α-SMA and the decrease in E-cadherin expressions induced by mannitol (200 mM) to the control level (Figs 6A and S2). Similar results were obtained for mRNA levels of the epithelial marker, E-cadherin, and the mesenchymal markers, α-SMA and vimentin (S3 Fig).

## Cotreatment with Y-27632 attenuates the hyperosmolarity-induced production of fibrogenesis-related factors

The differentiation of tubular epithelial cells into α-SMA-positive myofibroblasts ultimately leads to the deposition of the ECM, which contributes to the progression of renal fibrosis [6]. Given that hyperosmotic stress induced EMT in NRK-52E cells, we reasoned that the expression of ECM-related genes is promoted in response to hyperosmotic stress. To test this hypothesis, we first investigated whether hyperosmotic stress affects the expression of ECM-related genes using real-time PCR. Upon treatment with 200 mM mannitol for 24 h, the expression

levels of the ECM marker collagen I and the fibrogenesis factor PAI-1 were increased, but the increase in the expression of ECM fibronectin was not statistically significant (Fig 8). Since PAI-1 is a major inhibitor of plasminogen activator, increased PAI-1 levels are considered to contribute to increased ECM accumulation due to the suppression of plasmin production and matrix metalloproteinase activation [6]. Therefore, these results indicate that hyperosmotic stress induces the expression of fibrosis-related factors through EMT in NRK-52E cells. Moreover, treatment with Y-27632 abolished the upregulation of the mRNA levels of collagen I, PAI-1, and fibronectin in the presence of mannitol (Fig 8). These results suggest that in NRK-52E cells, the ROCK signaling inhibition attenuates the hyperosmotic stress-induced upregulation of profibrotic factors, and prevention of FAs rearrangements may be involved in part of the mechanism.

## Discussion

Our primary purpose in this study was to explore the role of hyperosmolarity in the EMT of proximal tubular epithelial cells. The novel finding of this study was that hyperosmotic mannitol stress could induce EMT, and FAs rearrangements are thought to be partly involved in the mechanism for the hyperosmotic stress-induced EMT.

Considering that using urea, which is a membrane-permeable osmolarity regulator did not induce EMT, the mannitol-induced EMT could be triggered by cell shrinkage due to osmotic differences between the cytosol and extracellular compartment and not by the hyperosmotic condition itself (Figs 1 and 3). We observed a decrease in projected cell areas after the mannitol treatment (Fig 4A), which suggests that a reduction of cell-cell contact occurred. Previous studies have demonstrated that disrupting the cell-cell contact induces the proteolytic shedding of E-cadherin, which causes the nuclear translocation of β-catenin, the transcriptional induction of Snail/Slug, and the repression of E-cadherin transcription in NRK-52E cells [43, 44]. We observed in this study that mannitol treatment of NRK-52E cells reduced E-cadherin expression (Fig 1) and increased the expression levels of Snail and Twist (Fig 2). Although we need further investigation the relationship between the repression of E-cadherin and the up-

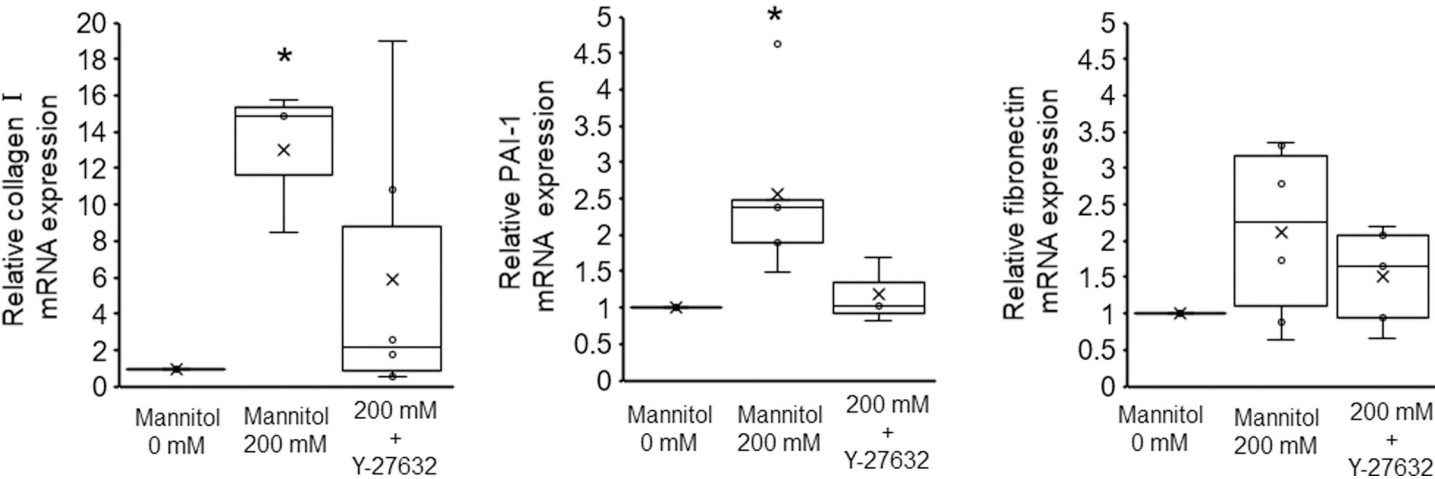

**Fig 8. Effects of Y-27632 on the hyperosmotic mannitol stress-induced expression of ECM-related genes.** NRK-52E cells were cotreated with mannitol (200 mM) and Y-27632 (1 μM), and mRNA expression was analyzed by real-time PCR. Quantitation of the changes in collagen-I (n = 6 from 0 mM, n = 3 from 200 mM, n = 6 from 200 mM + Y-27632), PAI-1 (n = 5 from 0 mM, n = 5 from 200 mM, n = 3 from 200 mM + Y-27632) and fibronectin (n = 6 from 0 mM, n = 6 from 200 mM, n = 5 from 200 mM + Y-27632). Relative gene expression levels were calculated considering mannitol (0 mM) as 1 and plotted. Data are presented as box and whisker plots with average (×), median, IQR and minimum and maximum values. The n indicates the number of independent experiments. *$P < 0.05$ from the data of 0 h (Steel–Dwass test).

regulation of Snail/Twist in response to hyperosmotic mannitol, it is reasonable to suppose that the decrease of E-cadherin, followed by the induction of EMT, in hyperosmotic mannitol stress was due to disrupting the cell-cell contact caused by cell shrinkage. To further elucidate the mechanism of hyperosmotic mannitol-induced EMT, it is important to investigate the effects on not only E-cadherin but also various epithelial cell markers, such as zonula occulu-dens-1 and N-cadherin which is thought to be the predominant classic cadherin in the proximal tubule *in vivo* [45–47].

Previous studies have demonstrated that hyperosmotic stress affects cytoskeletal structures such as actin fibers and microtubules [48, 49]. We also observed changes in actin cytoskeletal structure and the arrangements of FAs induced under mannitol-mediated hyperosmotic conditions. It is possible that hyperosmotic conditions affect syntheses of these proteins [50], which may cause the changes. On the other hand, a previous study demonstrated that hyperosmotic stress even for 10 min activates Rho family small GTPases, Rac/Cdc42, which contribute to volume-dependent cytoskeleton remodeling characterized by disassembly of stress fibers and deposition of peripheral actin filaments [29]. It is also reported that hyperosmotic stress activates Rho/ROCK and shrinkage-induced cofilin phosphorylation, which induces the reorganization of the actin cytoskeleton upon osmotic stress [51]. These results are quite similar to the changes in the actin cytoskeleton in our study, and our osmotic concentration of 200 mM mannitol (530 mOsmol/L) sufficiently exceeded the threshold for the activation of the ROCK signaling pathway shown by the previous study [52]. Thus, we think that the actin filaments were disassembled by ROCK signaling pathway under the 200 mM mannitol condition. Since vinculin was observed mostly at the end of actin stress fibers, and FAs act as sites of actin polymerization associating with the reorganization of actin cytoskeletal structure, we also suppose that hyperosmotic stress induces the disassembly and subsequent rearrangements of FAs concomitant with actin filament dynamics change.

We found that even the cell area and the structure of actin cytoskeletons recovered to the initial level under mannitol-mediated hyperosmotic condition, the number and the area of FAs increased at later time points (Figs 4A and 5). The mechanisms for these cell phenomena are still unclear, but we hypothesize a compensatory response of cells causes this. Instead of the decreased E-cadherin expression under hyperosmotic mannitol stress condition, cells may increase FAs to maintain intracellular force balance [53, 54], known as mechanical homeostasis, by remodeling mechanical coupling between cell-cell and cell-substrate adhesion, which is also suggested to play an important role in EMT [55].

Previous studies have reported that incorporation of α-SMA into stress fibers induces increases in the contractile activity of stress fibers and the FA size [30, 34]. It has also been reported that the contractile activity correlates with the expression level of α-SMA [32]. Our data revealed that the application of hyperosmotic stress induced the increased expression of α-SMA, which is more likely to be incorporated into stress fibers (Fig 6), as well as the increased size of FA (Fig 5D). These findings provide a strong indication for a hyperosmotic stress-induced positive feedback loop between α-SMA and FAs. The existence of such a feedback loop further supported by our findings that cotreatment with a ROCK inhibiter Y-27632 (1 μM), which is known to attenuate the contractile activity of stress fibers based on inhibition of ROCK and MLCK pathways [56, 57], inhibited the rearrangement of FAs, α-SMA expression, and recruitment of α-SMA to the actin stress fibers (Fig 6). Nevertheless, the Rho/ROCK signal also acts on other cytoskeletons, such as tubulin and intermediate filaments and regulates intracellular contractile forces [58–60]. Therefore, one of the limitations of our study is that we could not rule out the effects of Y-27632 on the contractile forces derived from other cytoskeletons except the actin cytoskeleton. However, our results are potentially important as they demonstrate that the hyperosmolarity-induced cytoskeletal changes may trigger the

differentiation of α-SMA-positive myofibroblasts. Investigating the effects of hyperosmolarity on the changes in intracellular contraction forces can further elucidate the mechanism underlying the effects of cytoskeletal changes on EMT.

The importance of EMT in the progression of renal fibrosis has been controversial [7, 61]. A previous study has reported that ∼5% of the total interstitial α-SMA-positive myofibroblasts arose from an EMT in unilateral ureteral obstruction model [7]. Thus, generation of α-SMA-positive myofibroblasts as a consequence of EMT in renal epithelial cells reflects only part of the biological processes of differentiation. However, our results are believed to be important because recent studies have disclosed that "partial EMT," in which epithelial cells remain attached to the tubular basement membrane but the epithelial cell transformation occurs, plays a vital role in initiating tubular dysfunction and driving fibrosis development [62–65]. It is known that partial EMT causes myofibroblast proliferation, triggering the cell cycle arrest of epithelial cells and promoting the release of fibrogenesis factors [63]. In fact, it has been shown that in animal models of renal fibrosis, several tubular cells are arrested in the cell cycle, and these tubular cells lead to the synthesis and secretion of profibrotic factors through partial EMT [63, 66]. In the present study, we found that mannitol treatment of NRK-52E cells increased the expression levels of Snail and Twist, two key transcription factors that regulate partial EMT (Fig 2), and collagen I and PAI-1, major profibrotic factors (Fig 8). Considering the importance of α-SMA-positive myofibroblasts in the pathogenesis of renal disease, our finding that hyperosmotic stress-induced the upregulation of partial EMT markers could have significant implications.

In summary, it is possible that the FA rearrangement in response to hyperosmotic mannitol is one of the mechanisms responsible for the EMT of proximal tubular epithelial cells. Our findings indicate the possibility that hyperosmotic stress, which generates mechanical stress, is a potential risk factor affecting the induction of EMT in proximal tubular cells.

## Supporting information

**S1 Fig. Effects of hyperosmolarity on actin cytoskeleton of NRK-52E cells.** Cells were cultured with 200 mM mannitol or urea for 0, 0.25, 0.5, 2, and 12 h. Typical fluorescence images of F-actin. Bar, 25 μm.
(TIF)

**S2 Fig. Effects of Y-27632 on the epithelial–mesenchymal transition of NRK-52E cells.** (A) Typical fluorescence images of E-cadherin (green), α-SMA (red), and combined (green and red) cotreated with mannitol (200 mM) and Y-27632 (1 μM) for 12 h. Bar, 25 μm. (B) Quantitation of the changes in the mean fluorescence intensity of E-cadherin and α-SMA (n = 34 from 200 mM + Y-27632) by immunofluorescence staining. The data of mannitol (0 and 200 mM) were identical to those in Fig 1B and 1C, which were shown for comparisons. Data are presented as box and whisker plots with average (×), median, IQR, and minimum and maximum values. The n indicates the number of cells analyzed. **P < 0.01 from the data of 0 mM (Tukey's test).
(TIF)

**S3 Fig. Effects of Y-27632 on the hyperosmotic mannitol stress-induced expression of EMT-related genes.** NRK-52E cells were cotreated with mannitol (200 mM) and Y-27632 (1 μM), and mRNA expression was analyzed by real-time PCR. Quantitation of the changes in E-cadherin (n = 3 from 0 mM, n = 3 from 200 mM, n = 3 from 200 mM + Y-27632), vimentin (n = 3 from 0 mM, n = 3 from 200 mM, n = 3 from 200 mM + Y-27632) and α-SMA (n = 3 from 0 mM, n = 3 from 200 mM, n = 3 from 200 mM + Y-27632). Relative gene expression

levels were calculated considering mannitol (0 mM) as 1 and plotted. Data are presented as box and whisker plots with average (×), median, IQR and minimum and maximum values. The n indicates the number of independent experiments.
(TIF)

**S1 Raw images. Raw images of western blot.** Raw images of western blot using in the Fig 6A.
(PDF)

**S1 File. Supporting data excel file.** All data that make up the figures within this paper is stored in this file.
(XLSX)

## Acknowledgments

We thank Drs. Nobuharu Fujii and Yasuko Manabe (Department of Health Promotion Sciences, Graduate School of Human Health Sciences, Tokyo Metropolitan University) and Dr. Kanae Ando (Department of Biological Sciences, School of Science, Tokyo Metropolitan University) for the biological technical supports (quantitative real-time PCR and Western blotting). We also thank Yuki Taninaka (Tokyo Metropolitan University) for technical supports.

## Author Contributions

**Conceptualization:** Takashi Miyano.

**Data curation:** Takashi Miyano, Atsushi Suzuki.

**Formal analysis:** Takashi Miyano.

**Funding acquisition:** Naoya Sakamoto.

**Project administration:** Naoya Sakamoto.

**Supervision:** Naoya Sakamoto.

**Writing – original draft:** Takashi Miyano, Naoya Sakamoto.

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
