## [Decision Letter · Decision Letter 0]

15 Sep 2021

PONE-D-21-23779Hyperosmotic Stress Induces Epithel­­ial-Mesenchymal Transition through Rearrangements of Focal Adhesions in Tubular Epithelial CellsPLOS ONE

Dear Dr. Miyano,

Thank you for submitting your manuscript to PLOS ONE. After careful consideration, we feel that it has merit but does not fully meet PLOS ONE’s publication criteria as it currently stands. Therefore, we invite you to submit a revised version of the manuscript that addresses the points raised during the review process.

Reviewer #1 has a number of substantial concerns and questions that should be addressed and clarified point by point, particularly the call to adopt a more "quantitative and consistent approach to evaluating EMT".  

We look forward to receiving your revised manuscript.

Kind regards,

Michael Klymkowsky, Ph.D.

Academic Editor

PLOS ONE

 “This study was supported in part by Grants-in-Aid for Scientific Research by the MEXT of Japan (No. 17H0277, 18H03521, and 18K19934).”          

Reviewers' comments:

Reviewer's Responses to Questions

**Comments to the Author**

1. Is the manuscript technically sound, and do the data support the conclusions?

Reviewer #1: No

Reviewer #2: Yes

2. Has the statistical analysis been performed appropriately and rigorously? 

Reviewer #1: Yes

Reviewer #2: Yes

3. Have the authors made all data underlying the findings in their manuscript fully available?

Reviewer #1: Yes

Reviewer #2: Yes

4. Is the manuscript presented in an intelligible fashion and written in standard English?

Reviewer #1: Yes

Reviewer #2: Yes

5. Review Comments to the Author

Reviewer #1: The current work showed the effect of hyperosmotic stimuli on EMT through Rho-ROCK mediated reorganization of cytoskeleton in rat tubular epithelial cell line, NRK-52E. EMT was evaluated and semi-quantitated mainly by immunostaining. Though several previous reports have already shown hyperosmotic stimuli induced the activation of Rho kinase, the concept that this phenomenon extends toward EMT process is novel.

However, there are several concerns for this work.

Major

1. One criticism for this work is lacking a more quantitative and consistent approach to evaluating EMT. Some experiments showed aSMA fluorescence intensity, but some showed SMA protein expression by western blot. For other mesenchymal molecules, Snail and Twist RNA expressions were assessed in some experiments, and collagen RNA expressions in others. This inconsistency of readouts throughout the manuscript is confusing. Authors should analyze the same or similar readouts even in different stimuli to cells.

2 EMT is defined as losing epithelial characteristics and acquiring the mesenchymal phenotypes. Authors only analyzed the loss of E-cadherin fluorescence intensity, but it is insufficient for showing the dynamics of epithelial markers. More quantitative analysis such as qPCR or western blot for various epithelial markers is necessary.

3 In introduction and discussion sections, authors argued the importance of EMT process and SMA+ epithelial cells on development of extracellular matrix producing interstitial myofibroblasts in injured kidney. However, recent lineage tracing analyses have clearly demonstrated that epithelial cells rarely ("never" in some papers) transdifferentiate into interstitial SMA+ myofibroblasts in vivo. Authors have to describe that aSMA expression as a consequence of EMT in renal epithelial cells is observed mainly in the cultured condition as limitation of this work.

Reviewer #2: Minor points:

All figures with immunofluorescence should be larger in diameter. Details are not clearly visible in these small images.

I would suggest to prepare better quality of Figures, by increasing dimensions of IF images.

6. PLOS authors have the option to publish the peer review history of their article (what does this mean?). If published, this will include your full peer review and any attached files.

Reviewer #1: No

Reviewer #2: No

---

## [Author Response · Author response to Decision Letter 0]

27 Oct 2021

We are grateful for the opportunity to revise our manuscript PONE-D-21-23779, and the helpful comments of your reviewers. All comments by reviewers have been addressed, with corresponding changes made directly to the manuscript where appropriate.

Responses to Reviewer #1 comments:

We would like to thank the reviewer for the detailed review and insightful comments. We have made every effort to revise the manuscript based on the recommendations and suggestions provided by reviewers. Revised texts are displayed in red with underlines in the manuscript. 

Major points:

1. One criticism for this work is lacking a more quantitative and consistent approach to evaluating EMT. Some experiments showed aSMA fluorescence intensity, but some showed SMA protein expression by western blot. For other mesenchymal molecules, Snail and Twist RNA expressions were assessed in some experiments, and collagen RNA expressions in others. This inconsistency of readouts throughout the manuscript is confusing. Authors should analyze the same or similar readouts even in different stimuli to cells.

Response: 

We thank the reviewer for this insightful comment. We used immunofluorescence imaging to assess the relationships between the morphological transition of cells and expression and dynamics of E-cadherin and α-SMA in response to hyperosmotic stress, but we agree with the reviewer comment, the only measurement of fluorescent intensity is less quantitative. We add Western blotting results of E-cadherin expression in the revised manuscript. We confirmed that the protein expression levels of E-cadherin and α-SMA were consistent with fluorescent intensities. We also showed that E-cadherin expression was decreased in response to 200 mM mannitol, and cotreatment with Y-27632 recovered the expression to the control level (new Fig 6). We revised the texts related to the results. (MATERIALS AND METHODS, Western blotting, p10; RESULTS, lines 312-313, p20; RESULTS, lines 370-373, p23)

As mentioned in the manuscript, Snail and Twist are well-known upstream transcription factors regulating E-cadherin expression [Ref. #1-3]. To verify that the decreased expression of E-cadherin was caused by hyperosmotic stress-induced cellular responses, we assessed the changes in these factors by qPCR. To confirm whether the hyperosmotic stress-induced EMT has a potential for kidney fibrosis, we also evaluated changes in mRNA expression levels of collagen, fibronectin, and PAI-1. We added these explanations in the revised manuscript (RESULTS, lines 203-205, p13; RESULTS, lines 379-381, p24)

2. EMT is defined as losing epithelial characteristics and acquiring the mesenchymal phenotypes. Authors only analyzed the loss of E-cadherin fluorescence intensity, but it is insufficient for showing the dynamics of epithelial markers. More quantitative analysis such as qPCR or western blot for various epithelial markers is necessary.

Response: 

We appreciate the reviewer's constructive comment. As a quantitative analysis, we added the qPCR result of E-cadherin in the revised manuscript. Other epithelial markers, such as zonula occludens-1 (ZO-1) and N-cadherin, are also present in the proximal tubule [Ref. #4], and coordinately impact on EMT in tubular epithelial cells. But the loss of cell-cell adhesion caused by EMT accompanies decreases of these intercellular adhesion proteins. In fact, previous studies have shown that both E-cadherin and ZO-1 epithelial expression decrease during EMT in response to various stimuli in NRK-52E [Ref. #5, 6]. Hence, we also added results of expression of α-SMA and vimentin as the mesenchymal markers, to further confirm EMT of tubular epithelial cells. The expression level of E-cadherin was decreased in response to 200 mM mannitol, and cotreatment with Y-27632 recovered the expression to the control level. In contrast, the expression of both α-SMA and vimentin increased in response to 200 mM mannitol, and cotreatment with Y-27632 suppressed the expression levels compared to the 200 mM mannitol alone. (S3 Fig). We revised the texts related to the reviewer's suggestion (MATERIALS AND METHODS, RNA extraction and Quantitative real-time PCR, p8; RESULTS, lines 373-375, p23).

In accordance with the Reviewer's suggestion, we clarified the discussion regarding this in the revised manuscript. (2nd paragraph, line 428-432, p27)

To further elucidate the mechanism of hyperosmotic mannitol-induced EMT, it is important to investigate the effects on not only E-cadherin but also various epithelial cell markers, such as zonula occuludens-1 and N-cadherin which is thought to be the predominant classic cadherin in the proximal tubule in vivo [45-47].

3. In introduction and discussion sections, authors argued the importance of EMT process and SMA+ epithelial cells on development of extracellular matrix producing interstitial myofibroblasts in injured kidney. However, recent lineage tracing analyses have clearly demonstrated that epithelial cells rarely ("never" in some papers) transdifferentiate into interstitial SMA+ myofibroblasts in vivo. Authors have to describe that aSMA expression as a consequence of EMT in renal epithelial cells is observed mainly in the cultured condition as limitation of this work.

Response: 

We agree with the reviewer's comment that epithelial cells rarely transdifferentiate into interstitial α-SMA myofibroblasts in vivo. A previous study has reported that a very small number of interstitial α-SMA myofibroblasts arose from EMT, accounting for ∼5% of the total interstitial α-SMA myofibroblasts in the unilateral ureteral obstruction (UUO) model [Ref. #7]. However, partial EMT, in which epithelial cells remain attached to the tubular basement membrane but the epithelial cell transformation occurs, plays a vital role in initiating tubular dysfunction and driving fibrosis development in vivo [Ref. #8-11]. In addition, many inhibitors and small molecules against EMT exerted profound therapeutic effects on renal interstitial fibrosis by suppressing differentiation into myofibroblasts and production of extracellular matrix [Ref. #12, 13]. Therefore, we believe that although the proportion of EMT-derived myofibroblasts is small, the EMT program potentially plays an important role in the progression of renal disease, which raises critical results in a cultured condition worthy of further investigation.

We added explanations in the Introduction section of the revised manuscript (1st paragraph, Lane 46-52, p3):

Epithelial-mesenchymal transition (EMT) is a widely accepted mechanism by which injured tubular epithelial cells transform into myofibroblasts, and is involved in the pathogenesis of not only chronic kidney diseases but also in acute kidney injury (AKI) [1, 2]. During EMT, the tubular epithelial cells lose their epithelial characteristics and acquire mesenchymal features, concomitant with the downregulation of epithelial markers, including E-cadherin, and the upregulation of mesenchymal markers, including α-smooth muscle actin (α-SMA) and vimentin [3]. α-SMA-positive myofibroblasts are known to induce the expression of profibrotic factors such as collagen, fibronectin, and plasminogen activator inhibitor type 1 (PAI-1) [4-6]. In fact, a very small number of interstitial α-SMA-positive myofibroblasts has been shown to arise from an EMT in vivo model [7]. However, many inhibitors and small molecules suppressing EMT exerted profound therapeutic effects on differentiation into myofibroblasts and production of extracellular matrix [8, 9]. Therefore, although the proportion of EMT-derived myofibroblasts is small, the EMT program is believed to play an important role in the progression of renal disease.

To address the Reviewer's comment, we updated the Discussion section of the revised manuscript (6th paragraph, p481-485, p30) 

The importance of EMT in the progression of renal fibrosis has been controversial [7, 61]. A previous study has reported that ∼5% of the total interstitial α-SMA-positive myofibroblasts arose from an EMT in unilateral ureteral obstruction model [7]. Thus, generation of α-SMA-positive myofibroblasts as a consequence of EMT in renal epithelial cells reflects only part of the biological processes of differentiation. However, our results are believed to be important because recent studies have disclosed that “partial EMT,” in which epithelial cells remain attached to the tubular basement membrane but the epithelial cell transformation occurs, plays a vital role in initiating tubular dysfunction and driving fibrosis development [63-66].

In accordance with the Reviewer’s comment, we added the following references

#1 Lamouille S, Xu J, Derynck R. Molecular mechanisms of epithelial-mesenchymal transition. Nat Rev Mol Cell Biol 15: 178-196, 2014.

#2 Cano A, Pérez-Moreno MA, Rodrigo I, Locascio A, Blanco MJ, del Barrio MG, et al. The transcription factor snail controls epithelial-mesenchymal transitions by repressing E-cadherin expression. Nat Cell Biol 2: 76-83, 2000.

#3 Conacci-Sorrell M, Simcha, Ben-Yedidia T, Blechman J, Savagner P, Ben-Ze'ev A. Autoregulation of E-cadherin expression by cadherin-cadherin interactions: the roles of beta-catenin signaling, Slug, and MAPK. J Cell Biol 163: 847-857, 2003.

#4 Bolati D, Shimizu H, Higashiyama Y, Nishijima F, Niwa T. Indoxyl sulfate induces epithelial-to-mesenchymal transition in rat kidneys and human proximal tubular cells. Am J Nephrol 34: 318–323, 2011.

#5 Zhao L, Li C, Zhou B, Luo C, Wang Y, Che L, et al. Crucial role of serum response factor in renal tubular epithelial cell epithelial-mesenchymal transition in hyperuricemic nephropathy. Aging (Albany NY) 11(22):10597-10609, 2019.

#6 Bai Y, Lu H, Hu L, Hong D, Ding L, Chen B. Effect of Sedum sarmentosum BUNGE extract on aristolochic acid-induced renal tubular epithelial cell injury. J Pharmacol Sci 124:445-456, 2014.

#7 LeBleu VS, Taduri G, O’Connell J, Teng Y, Cooke VG, Woda C, et al. Origin and function of myofibroblasts in kidney fibrosis. Nat Med 19:1047-1053, 2013.

#8 Grande MT, Sánchez-Laorden B, López-Blau C, De Frutos CA, Boutet A, Arévalo M, et al. Snail1-induced partial epithelial-to-mesenchymal transition drives renal fibrosis in mice and can be targeted to reverse established disease. Nat Med 21:989-997, 2015.

#9 Lovisa S, LeBleu VS, Tampe B, Sugimoto H, Vadnagara K, Carstens JL, et al. Epithelial-to-mesenchymal transition induces cell cycle arrest and parenchymal damage in renal fibrosis. Nat Med 21:998-1009, 2015.

#10 Ovadya Y, Krizhanovsky V. A new Twist in kidney fibrosis. Nat Med 21:975-977, 2015.

#11 Zhou D, Liu Y. Renal fibrosis in 2015: Understanding the mechanisms of kidney fibrosis. Nature Reviews Nephrology 12:68-70, 2016.

#12 Ruiz-Ortega M, Rayego-Mateos S, Lamas S, Ortiz A, Rodrigues-Diez RR. Targeting the progression of chronic kidney disease. Nat Rev Nephrol 16: 269-288, 2020.

#13 Feng YL, Wang WB, Ning Y, Chen H, Liu P. Small molecules against the origin and activation of myofibroblast for renal interstitial fibrosis therapy. Biomed Pharmacother 139: 111386, 2021.

Responses to Reviewer #2 comments:

We would like to thank the reviewer for the detailed review. We have made every effort to revise the manuscript based on the recommendations provided by reviewers. 

Minor points:

All figures with immunofluorescence should be larger in diameter. Details are not clearly visible in these small images. I would suggest to prepare better quality of Figures, by increasing dimensions of IF images.

Response: 

We appreciate the reviewer's constructive comment. We modified the corresponding figures to be as large in diameter as possible.

---

## [Decision Letter · Decision Letter 1]

1 Dec 2021

Hyperosmotic Stress Induces Epithel­­ial-Mesenchymal Transition through Rearrangements of Focal Adhesions in Tubular Epithelial Cells

PONE-D-21-23779R1

Dear Dr. Miyano,

We’re pleased to inform you that your manuscript has been judged scientifically suitable for publication and will be formally accepted for publication once it meets all outstanding technical requirements.

Kind regards,

Michael Klymkowsky, Ph.D.

Academic Editor

PLOS ONE

Additional Editor Comments (optional):

Reviewers' comments:

Reviewer's Responses to Questions

**Comments to the Author**

1. If the authors have adequately addressed your comments raised in a previous round of review and you feel that this manuscript is now acceptable for publication, you may indicate that here to bypass the “Comments to the Author” section, enter your conflict of interest statement in the “Confidential to Editor” section, and submit your "Accept" recommendation.

Reviewer #1: All comments have been addressed

2. Is the manuscript technically sound, and do the data support the conclusions?

Reviewer #1: Yes

3. Has the statistical analysis been performed appropriately and rigorously? 

Reviewer #1: Yes

4. Have the authors made all data underlying the findings in their manuscript fully available?

Reviewer #1: Yes

5. Is the manuscript presented in an intelligible fashion and written in standard English?

Reviewer #1: Yes

6. Review Comments to the Author

Reviewer #1: (No Response)

7. PLOS authors have the option to publish the peer review history of their article (what does this mean?). If published, this will include your full peer review and any attached files.

Reviewer #1: No

---

## [Editor Report · Acceptance letter]

10 Dec 2021

PONE-D-21-23779R1 

Hyperosmotic Stress Induces Epithel­­ial-Mesenchymal Transition through Rearrangements of Focal Adhesions in Tubular Epithelial Cells 

Dear Dr. Miyano:

I'm pleased to inform you that your manuscript has been deemed suitable for publication in PLOS ONE. Congratulations! Your manuscript is now with our production department. 

Kind regards, 

on behalf of

Dr. Michael Klymkowsky 

Academic Editor

PLOS ONE